# Liver Regeneration as a Model for Studying Cellular Plasticity in Mammals: The Roles of Hepatocytes and Cholangiocytes

**DOI:** 10.3390/cells14151129

**Published:** 2025-07-22

**Authors:** Andrey Elchaninov, Polina Vishnyakova, Valeria Glinkina, Timur Fatkhudinov, Gennady Sukhikh

**Affiliations:** 1Laboratory of Growth and Development, Avtsyn Research Institute of Human Morphology of FSBI “Petrovsky National Research Centre of Surgery”, 3 Tsurupa Street, 117418 Moscow, Russia; tfat@yandex.ru; 2Laboratory of Molecular Pathophysiology, Research Institute of Molecular and Cellular Medicine, Peoples’ Friendship University of Russia (RUDN University), 6 Miklukho-Maklaya Street, 117198 Moscow, Russia; vpa2002@mail.ru (P.V.); vglinkina@mail.ru (V.G.); 3Laboratory of Regenerative Medicine, Institute of Translational Medicine, National Medical Research Centre for Obstetrics, Gynecology and Perinatology Named After Academician V.I. Kulakov of Ministry of Healthcare of Russian Federation, 4 Oparina Street, 117997 Moscow, Russia; gt_sukhikh@bk.ru

**Keywords:** liver, regeneration, hepatocytes, cholangiocytes, cell plasticity

## Abstract

In most countries, liver disease is one of the most common pathologic conditions among the population. In this regard, the development of new methods to treat liver diseases is not possible without understanding the mechanisms of regeneration of this organ. A characteristic reaction of the liver to certain damaging factors is a pronounced cellular plasticity; this primarily concerns hepatocytes and cholangiocytes. This property is also characteristic of Ito stellate cells and macrophages. In this study, we focus on the plasticity of hepatocytes and cholangiocytes. We consider such manifestations of plasticity as the ability to enter the mitotic cycle, as well as transdifferentiation. The contribution of each type of plasticity to liver regeneration is considered, as well as the molecular mechanisms providing the cellular plasticity of hepatocytes and cholangiocytes.

## 1. Introduction

Phenotype plasticity is inherent in all cells, albeit to varying degrees, that adapt to changing conditions of functioning. From biological and medical perspectives, the most important manifestations of cellular plasticity are the abilities to proliferate and transdifferentiate. Obviously, both of these phenomena underlie the formation of tumors. However, the ability to proliferate and transdifferentiate is also related to physiological and reparative regeneration [1]. The contribution of each of these processes to regeneration varies significantly among different representatives of the animal phyla, as well as depending on the nature of the damaging factor [2].

According to current concepts, the degree of cellular plasticity in the phylogeny of many taxonomic groups has decreased [2]. This decreased cellular plasticity, in the opinion of many researchers, underlies the decrease in the ability to perform reparative regeneration. This is especially noticeable in the case of mammals and the regulation of the mitotic cycle in their cells [3,4].

The liver is one of the few mammalian organs that exhibits pronounced cellular plasticity under normal conditions and during regeneration [5]. In this literature review, we will consider such manifestations of cellular plasticity as the ability of liver cell types, particularly hepatocytes and cholangiocytes, to proliferate and transdifferentiate (Figure 1 and Figure 2). We will also consider the contribution of these processes to maintaining tissue homeostasis in the liver under normal conditions, as well as during regeneration after various types of damage. This review intentionally does not cover the issue of the phenotypic plasticity of other liver cell types, including Kupffer cells, since this issue has been discussed in detail by the authors elsewhere [6,7,8,9].

It is worth dwelling on the terms used. Cellular plasticity is a rather complex concept and can be considered in at least two aspects: (1) a change in the functional status of a particular cell and (2) the transformation of one cell type into another. The transition of hepatocytes from the G0 to G1 stage of the mitotic cycle can be an example of the first variant of cellular plasticity. Within the framework of the second concept, there is the term transdifferentiation. The classical variant of transdifferentiation assumes the transformation between cells of different germ layers. This is observed during regeneration in some invertebrates [2]. In a number of works, transdifferentiation is considered more broadly as any transformation of some cell types into others, including within one germ layer. In the presented work, we use the term transdifferentiation in a broad sense. An example is the transformation of cholangiocytes into hepatocytes and vice versa. In this case, both cell types, as is known, develop from one germ layer—the endoderm. When discussing the term transdifferentiation, the concept of dedifferentiation is often mentioned as an intermediate stage between two extreme cell types. There is no established definition of the term dedifferentiation. It is generally accepted that dedifferentiation is a certain simplification in the structure of a particular cell.

## 2. Hepatocytes

### 2.1. Major Signaling Cascades Activating Hepatocyte Proliferation

Hepatocytes constitute over 80% of the liver mass [10]. Under normal conditions, almost all hepatocytes are arrested in the G0 phase of the mitotic cycle. Mitotically dividing hepatocytes, which can be detected with a frequency of 1 in 20,000 cells, are believed to be located predominantly in the intermediate zone of the classical lobule (between the pericentral and periportal zones) [11]. The resection of the liver induces a wave of proliferation in hepatocytes (Figure 1). The minimum resection volume of parenchyma that is capable of causing a significant increase in the number of mitotically dividing liver cells in rats and mice is estimated to be 10% [12,13]. The most pronounced proliferation of hepatocytes is observed after 70% liver resection. Under these conditions, almost all hepatocytes enter the mitotic cycle and undergo 1–2 cell divisions [14,15]. In connection with this rapid transition from performing multiple metabolic functions to proliferation, hepatocytes have been dubbed as “phenotypic acrobats” [14,16].

The resection of the liver causes changes in the expression levels of multiple genes in hepatocytes. Several differentially expressed gene clusters were identified, including acute-phase response genes, proliferation genes, and cell adhesion genes. In addition to being distributed across different functional clusters, the differentially expressed genes were grouped according to a particular phase of liver regeneration after 70% resection as follows: initial phase (priming)—2–12 h; hepatocyte proliferation—24–36 h; and terminal phase—72–168 h post-resection. Thus, it was found that *Socs3*, *Lbp*, *Pap*, *Reg3a*, *A2m*, *Cxcr4*, *Mx1*, and *Mx2* were expressed among the acute-phase genes. Genes controlling hepatocyte proliferation are actively expressed in the middle phase of liver regeneration after 70% parenchyma resection; these include genes of various cyclins—*Ccna2*, *Ccnb1*, *Ccnb2*, *Ccnd1*, and *Ccne1*—as well as genes controlling karyo- and cytokinesis—*Cdc20*, *Cdc25b*, *Cdc2a*, *Cdca1*, *Cdca2*, *Cdca3*, *Cdca8*, *Kif11*, *Kif15*, *Kif22*, *Kif23*, *Kif2c*, and *Kifc1*. At the terminal stage of liver regeneration, hepatocytes actively express cell adhesion and extracellular matrix molecules represented by procollagen genes *Col14a1*, *1a1*, *1a2*, *3a1*, *4a1*, *4a2*, *5a1*, *5a2*, *6a2*, and *8a1*, as well as genes encoding cell junction proteins *gja4* and *gjb6* [17]. Also at the terminal stage of liver regeneration, many genes involved in the TGFβ/BMP signaling pathway are activated in hepatocytes [17,18].

Concerning the initial phase of liver regeneration (priming), special attention is paid to the role of TNFα and Il6, which regulate the entry of hepatocytes into the mitotic cycle, as well as HGF, EGF, and other factors, which stimulate hepatocytes to enter the mitotic cycle (Figure 1) [19,20].

According to research data, TNFα and IL6 are synthesized by liver macrophages and form a regulatory system that stimulates the expression of so-called early response genes in hepatocytes, entailing the entry of the cell into the mitotic cycle [20,21]. In particular, it has been shown that TNFα/IL6 stimulates the synthesis of iNOS-producing NO, which protects hepatocytes from the high levels of TNFα/IL 6, thereby preventing their death [22]. In addition, IL6 has recently been shown to induce the synthesis of the hepatocyte growth factor (HGF), which, in addition to stimulating the proliferation of epithelial cells and suppressing apoptosis, exerts an anti-inflammatory effect along with IL10 [23]. The mitotic cycling of hepatocytes is apparently ensured by HGF. In rats, the blood plasma levels of HGF increase more than 20-fold during the 1st hour after partial hepatectomy, remain at this level for 72 h, and then gradually return to normal values [19,21,24].

HGF is synthesized in the liver by stellate cells and, according to some reports, by sinusoidal endothelial cells [14,25]. The inactive form of HGF is found in large quantities in the extracellular matrix between hepatocytes, especially in the periportal region of the liver lobule [14]. Following liver injury, plasma urokinase levels increase, which triggers a cascade of proteolytic reactions that digest the extracellular matrix of the liver, thereby enabling the activation and release of HGF [19,24,26]. The main role in this process is performed by the family of matrix metalloproteinases [27,28].

Other growth factors that activate hepatocyte proliferation in the liver after resection include epidermal growth factor (EGF) and transforming growth factor-α (TGF-α) [29]. The removal of the salivary glands in rats causes a decrease in EGF blood levels, which leads to a decrease in the regenerative capacity of the liver. TFGα is synthesized by hepatocytes and has an autocrine effect. It is assumed that EGF acts in the early stages of liver regeneration, while TFGα acts in the later stages of the regenerative process, since the rate of TFGα synthesis reaches a peak only 24 h post-resection [14,19,24].

The action of the above-mentioned growth factors results in the activation of proto-oncogene and cyclin genes, the expression of which induces the cell to divide [19,21,24]. The role of microRNA in the regulation of hepatocyte proliferation has been extensively studied. MicroRNAs suppress gene expression at the translation level and/or by promoting mRNA degradation [25]. It has been shown that changes in the expression of miR-21, miR-221, miR-26a, and other microRNAs during liver regeneration correlate with the activation of genes encoding growth factors and cell cycle regulators [30,31,32].

The Wnt signaling pathway plays a prominent role in mammalian liver regeneration [14,33]. The ligands of this signaling pathway are synthesized in the liver mainly by Kupffer cells and endothelial cells [34,35]. The initial link of the cascade is β-catenin, which is a mediator for the activation of target genes *c-Myc* and *Cyclin D1*. β-catenin in the membrane can be phosphorylated by various kinases, including c-met, EGFR, etc., which leads to the separation of β-catenin from E-cadherin and its translocation to the nucleus, where it regulates the expression of Wnt-dependent genes c-*Myc* and *Cyclin D1* [14,33].

NOTCH signaling is another molecular cascade that maintains tissue homeostasis in the mammalian liver. The NOTCH pathway is one of the regulatory systems that controls the life cycle, differentiation, and proliferation of cells in most multicellular organisms. The NOTCH system includes four types of transmembrane NOTCH receptors (NOTCH-1, -2, -3, and -4), two SERRATE/JAGGED ligands (Jag-1 and -2), and three DELTA-LIKE ligands (Dll-1, -3, amd -4), as well as a number of intracellular proteins [36]. Four NOTCH receptor genes are expressed in the liver of adult mammals; *Notch1* and *Notch2* genes are predominantly expressed in hepatocytes and cholangiocytes, while *Notch3* and *Notch4* expression is observed in stromal cells [37].

There are few contradictory reports on the role of the NOTCH system in liver regeneration after resection. One study shows that the synthesis of NOTCH1 and JAGGED1 proteins increases in hepatocytes and cholangiocytes within 4 days after resection of 70% of the rat liver mass. The suppression of the corresponding genes by RNA interference causes a decrease in hepatocyte proliferation after liver resection [38]. However, another study demonstrates the spontaneous activation of hepatocyte proliferation and nodular hyperplasia without connective tissue proliferation in the non-operated liver of mice with the inactivated *Notch1* gene. In animals of the same line, after liver resection, the rate of regeneration and the level of hepatocyte proliferation were lower than in wild-type animals with normal *Notch1* expression [39]. This difference may reflect that by the time of surgery, the liver mass in animals with suppressed *Notch1* expression was already increased, hence the liver resection resulting in a smaller reduction in organ mass relative to the body mass of the animal [39]. Overall, NOTCH signaling is of key importance in the regulation of hepatocyte proliferation during liver regeneration after the partial removal of the organ.

Once the liver mass reaches its initial values during reparative regeneration after resection, the regeneration process is complete. There are many factors that stop liver regeneration in mammals. One of the first to be described was TGF-β1, which has been shown to inhibit proliferation in hepatocyte cultures [21,24].

### 2.2. Dependence of Hepatocyte Proliferation on Localization

As already noted, after 70% liver resection, almost all hepatocytes in the remnant organ enter proliferation [15]. However, given the presence of different zones with different levels of oxygenation and nutrition within the liver lobule [40], it is obvious that hepatocytes will enter the mitotic cycle at different rates. Classical studies have shown that periportal hepatocytes are the earliest to proliferate, followed by pericentral hepatocytes [13,41], which can be associated by their different sensitivities to mitogens; pericentral hepatocytes, for example, are less sensitive to HGF and EGF [42]. It is suggested that, in addition to the peculiarities of lobular blood supply, the Wnt signaling pathway is also involved in maintaining differences in the properties and function of hepatocytes within the periportal, intermediate, and pericentral zones [43]. It has been shown that pericentral hepatocytes are under the constant influence of the Wnt 2 and Wnt 9 proteins synthesized by endothelial cells of the liver sinusoids of the corresponding zone of the liver lobule [43].

The non-uniform entry of hepatocytes into proliferation can create the illusion of the presence of special subpopulations of hepatocytes in the liver that are predominantly involved in maintaining the number of hepatocytes during regeneration. At the same time, such hepatocytes exhibit some features of poorly differentiated cells [44]. There is no consensus regarding the localization of such hepatocytes within the liver lobule. According to some studies, such hepatocytes are periportal, since they express the Fn14 surface receptor for the TWEAK progenitor cell activation factor, as well as the SOX9 transcription factor [45,46]. In our studies using a rat liver regeneration model after subtotal resection (approximately 80% of the organ mass), we also observed the appearance of SOX9+ hepatocytes in significant numbers. However, such hepatocytes were not associated with the periportal zone, while hepatocytes with mitotic figures significantly exceeded the SOX9+ hepatocytes in number. After the removal of 80% of the liver mass in rats, a temporary block of the mitotic cycle is observed for 36 h after surgery; as such, we concluded that the activation of SOX9 synthesis in hepatocytes contributed to the exit from this block and the entry of hepatocytes into the mitotic cycle. This conclusion is based on data suggesting that SOX9 is synthesized in cells that are capable of active proliferation [47,48].

Other studies show that pericentral hepatocytes are the earliest to proliferate and provide a larger contribution to liver mass restoration upon resection. It has been shown that pericentral hepatocytes are mainly diploid and express the Tbx3 marker of poorly differentiated hepatocytes. The hepatocytes are maintained in this state by endothelial cells of the central vein, which produce Wnt signaling pathway proteins [49,50]. This concept of the cellular basis of liver regeneration differs from the traditional view, according to which, during liver regeneration after resection, virtually all hepatocytes in the liver lobule begin to proliferate. Ultimately, a more thorough examination of the role of so-called hepatocyte subpopulations in liver regeneration confirms the classical concepts [51].

According to several studies, dividing hepatocytes under normal conditions are found predominantly in the intermediate zone of the classical liver lobule. As a result, with age, the relative volume of the pericentral and periportal zones decreases, while that of the intermediate zone increases [52,53]. The data argue that the proliferating hepatocytes in the intermediate zone are virtually the only source of liver restoration after damage. However, this conclusion was made only on the basis of using models of toxic liver damage, in which hepatocytes of the periportal and/or pericentral zones predominantly die. Under conditions when only the intermediate zone remains intact, it is quite obvious that liver regeneration will be carried out by hepatocytes of the only preserved zone of the liver lobule. It is noteworthy that after partial hepatectomy, the same authors demonstrate different dynamics of hepatocyte proliferation, corresponding to classical concepts, with almost all hepatocytes being capable of entering the mitotic cycle. However, proliferation begins in the periportal zone; then, dividing hepatocytes appear in the intermediate zone and, finally, in the pericentral zone of the liver lobule [15,53].

### 2.3. Factors Influencing Hepatocyte Proliferation

The proliferation of hepatocytes during liver regeneration is influenced by multiple factors including age. In general, it can be noted that the younger the animal, the faster its hepatocytes enter the mitotic cycle. Thus, the mitotic activity of hepatocytes in 5-day-old rats increases after 20 h; in 4–6-week-old rats, it increases after 24 h; in 4–6-month-old rats, it increases after 48 h; and in 16-month-old rats, it increases after 72 h post-resection [13,54]. Variability in the latent time of proliferation and the time point of peak cell proliferation in the regenerating liver in animals of different age groups apparently depends on the expression levels of factors that suppress cell division. In rats, the synthesis of the C/EBPα proliferation inhibitor, which is found in fetal livers before birth, leads to a gradual decrease in the mitotic activity of hepatocytes with age [55]. In young animals, after partial hepatectomy, the expression levels of C/EBPα and the cyclin-dependent kinase inhibitor p21 are sharply reduced, which does not occur in old animals [56]. It is likely that the lower level of C/EBPα, as well as the rapid decrease in C/EBPα and p21 levels post-resection, determines the earlier onset of hepatocyte proliferation during liver regeneration in young individuals. For the same reason, in young animals, the peak of hepatocyte mitoses is observed after the removal of a smaller amount of liver parenchyma compared to adult individuals; in adult rats, it requires the removal of approximately 40% of the liver mass, while in rat pups in the early postnatal period, it requires the removal of 20% of the liver mass [12,57].

Thus, the sequence of events during liver regeneration can be generally represented as follows: After damage, the priming phase begins, during which hepatocytes are prepared to enter the mitotic cycle; then, under the influence of mitogens, each hepatocyte performs 1–2 mitotic divisions. Hepatocyte proliferation is followed by that of other liver cell types (cholangiocytes, macrophages, endotheliocytes, and Ito cells), after which the regeneration is complete.

### 2.4. Transdifferentiation Capacity of Hepatocytes

The ability to form tubular structures has long been known in hepatocytes. Initially, this phenomenon was described in hepatocyte cultures. At present, it is assumed that this ability in hepatocytes is also achieved in vivo under conditions of chronic liver damage, especially when cholangiocytes are most severely affected [58,59,60]. The cells that emerge from hepatocyte transdifferentiation are similar to the cells that emerge from cholangiocytes in the course of the ductular reaction (DR), which is characterized by liver damage-induced cell proliferation in the reactive bile ducts (Figure 1) [61]. However, unlike cholangiocyte derivatives, such cells are bigger and may contain two nuclei, as well as expressing the CK19 and Epcam genes at lower levels compared to ‘true’ cholangiocytes [62,63]. The heterogeneity of cholangiocytes formed in the DR was also confirmed using a single-cell analysis, revealing cells that express hepatocyte markers *Alb*, *Agt*, *Ttr*, *Apoe*, *Apoc1*, *Gstm1*, *Gsta4*, and *Cyp2j6* among the studied population of cells. The data confirm the ability of hepatocytes to differentiate into cholangiocyte-like cells [64].

Some studies distinguish two subpopulations of cells derived from hepatocytes. This distinction is based on the presence or absence of markers that are typical of cholangiocytes [65]. In fact, these two subpopulations represent the successive stages of differentiation from hepatocytes to cholangiocytes [66].

Previously, cells emerging from hepatocyte transdifferentiation were assumed to have a stable phenotype [59]. In later studies, data were obtained on the ability of cholangiocyte-like cells to return to the phenotype of typical hepatocytes in the absence of liver damage [62].

It is assumed that the pronounced cell plasticity of hepatocytes and cholangiocytes may underlie the pathogenesis of liver cancer. The YAP signaling pathway provides the key signaling that regulates the plasticity of liver epithelial cells during prenatal development, as well as during the regeneration and development of liver tumors (Figure 1) [67]. The transdifferentiation of hepatocytes into cholangiocyte-like cells is also a YAP-dependent process [64]. The expression of the SOX9 marker, which is a member of the YAP regulatory cascade, is a characteristic feature of cholangiocytes [68]. The activation of the YAP signaling pathway and SOX9 in hepatocytes has been implicated in the formation of extremely aggressive hepatocellular carcinoma [66,69].

In addition to NOTCH and YAP signaling, SOX9 expression can also be activated by IL6. In a DDC-injured liver model, Kupffer cells were shown to produce IL6, which induces the formation of pSTAT3. This, in turn, stimulates the expression of genes related to biliary reprogramming, particularly *Sox9* and *Spp1* [70].

The role of TGFβ signaling in hepatocyte reprogramming has been revealed. It has been established that the knockout of the *Smad4* gene promotes hepatocyte transdifferentiation toward cholangiocytes in the DDC-injured mouse liver model [71].

Recently, data have emerged on SOX9 expression following liver resection in mice [72]. It is interesting to note that in 2016, there was a sharp increase in the number of SOX9+ hepatocytes after subtotal liver resection (more than 80% of the organ mass) in rats, which was previously reported by us [47,73]. Given the ascertained role of the YAP signaling pathway and SOX9 in ensuring the plasticity of hepatocytes and other cells, it is likely that SOX9 expression in hepatocytes reflects their preparation for proliferation and deeper dedifferentiation in models of subtotal resection, after which a temporary block in the mitotic cycle of hepatocytes is observed.

In addition to SOX9, the role of SOX4 in the reprogramming of hepatocytes toward cholangiocytes has been demonstrated. In this context, the knockout of SOX9 prevented the appearance of all intermediate stages (early and late) between hepatocytes and cholangiocytes. The knockout of SOX4 inhibited the emergence of the very first cells with an intermediate phenotype, which still resembled hepatocytes [65].

At present, a study exists investigating the role of epigenetic factors involved in regulating the transdifferentiation of hepatocytes into cholangiocytes. It has been shown that the histone methyltransferase Nsd1 and the demethylase Kdm2a have reciprocal effects on H3K36 methylation, regulating the early and late stages of reprogramming, respectively [74]. In addition, it has been established that after the completion of hepatocyte-to-cholangiocyte transdifferentiation, the methylation profile of the cells remains virtually unchanged. According to the authors, this may explain why the emerging phenotype is not stable and why the cells eventually revert to the hepatocyte phenotype [75].

Some authors consider the plasticity of hepatocytes more broadly. Based on the common sources of embryonic development, a number of researchers suggest that it is possible to transdifferentiate hepatocytes into pancreatic islet cells for therapeutic purposes [76]. Interestingly, in CCl4 hepatotoxic injury models, hepatocytes show an increase in the expression of genes associated with both exocrine (trypsinogen-2, amylase-2, elastase-1, elastase-2, and cholesteryl ester lipase) and endocrine (Reg-1 and insulin genes) cells of the pancreas [77]. A pioneering study in the field of obtaining pancreatic islet cells using genetic engineering was conducted by Sarah Ferber, who used adenovirus to deliver the Pdx1 gene to hepatocytes, which resulted in insulin production by the transduced cells and a decrease in hyperglycemia [78]. In addition to Pdx1, other studies have proposed introducing other transcription factors, primarily Ngn3 and Tgif2 [79,80].

Evidence of transdifferentiation between hepatocytes and cholangiocytes was also found in a study of patients with liver steatosis. It was shown that the PI3K–AKT–mTOR pathway plays a key role in this process [81]. Particularly pronounced signs of transdifferentiation were observed in the final stages of the disease, where cells simultaneously positive for the markers ALB, CK7, and CK19 were detected in significant numbers [82].

## 3. Cholangiocytes

### 3.1. Major Signaling Cascades Activating Cholangiocyte Proliferation

Cholangiocytes, like hepatocytes, are dormant cells that proliferate at extremely low rates under normal conditions [82]. Cholangiocytes, like hepatocytes, are capable of entering the mitotic cycle upon liver damage (Figure 2) [14]. Cholangiocyte proliferation levels become significant only when the liver is damaged. Several types of cholangiocyte proliferation have been identified [83]. Typical cholangiocyte proliferation results in the growth of bile ducts with a well-defined lumen in the portal tract areas [82]. This type of proliferation is usually studied in models of the common bile duct ligation [84], 70% liver resection [85], acute liver injury with CCl4 [86,87], and the introduction of bile salts into the diet [88]. Certain heterogeneities of cholangiocytes within the biliary tract have been described [89,90].

A distinction is made between the so-called small and large cholangiocytes [81]. These types of cholangiocytes participate differently in typical proliferation. When the common bile duct is ligated, large cholangiocytes begin to proliferate [88]; after 70% liver resection, proliferation of both types of cholangiocytes is observed [85]; small cholangiocytes proliferate after acute injury with CCl4 [86]. The specific mechanisms that determine the entry of a particular type of cholangiocyte into proliferation are unclear.

The atypical proliferation of cholangiocytes is observed in patients with chronic cholestatic liver disease (primary sclerosing cholangitis and primary biliary cirrhosis) [91]. The pathology is characterized by the proliferation of cholangiocytes in the form of cords that spread among hepatocytes [91]. The multiplication of oval cells, which are regarded as the third variant of cholangiocyte proliferation, leads to the formation of cell cords with poorly defined lumen and often precedes liver tumorigenesis in laboratory rats [92,93]. The issue of oval cell formation in laboratory rodents is closely related to the ability of cholangiocytes to transdifferentiate.

Typical cholangiocyte proliferation has been studied in detail, with the common bile duct ligation model being the most commonly used model [94]. It is believed that cholangiocyte proliferation under conditions of common bile duct ligation is triggered by an increase in bile pressure in the ducts. However, the specific mechanisms remain poorly understood [83].

Several factors are considered to be major in this regard [95]. The proliferation of cholangiocytes has been shown to be preceded by the proliferation of endothelial cells of the capillaries surrounding the bile ducts [96]. Accordingly, an increase in VEGF synthesis can be one of the primary factors stimulating the proliferation of cholangiocytes [97]. The dependence of the typical proliferation of cholangiocytes on parasympathetic innervation has also been demonstrated [98,99]. At the same time, a total vagotomy or a block of dopaminergic innervation prevents bile duct hyperplasia under conditions of common bile duct ligation [98,99].

Somatostatin has a blocking effect on cholangiocyte proliferation, but this only applies to large cholangiocytes [82,100]. A blocking effect has also been shown for gastrin [101]. Bile acids that penetrate into cholangiocytes via the Na+-dependent apical bile acid transporter (ABAT) exert a stimulating effect on cholangiocyte proliferation [102]. The rapid removal of bile acids by duct drainage leads to a decrease in cholangiocyte hyperplasia [103].

The role of estrogens in the proliferation of the biliary epithelium has been ascertained [104]. In particular, cholangiocytes express estrogen receptors (ERs) alpha and beta, the expression of which increases under conditions of the common bile duct ligation. This finding is consistent with the data on the stimulating effect of 17-beta estradiol on hepatocyte proliferation and the increased incidence of primary biliary cirrhosis in women [105].

The mentioned studies on the influence of particular factors on cholangiocyte proliferation demonstrate the dependence of this process on the expression levels of secretin receptors, as well as on cAMP levels associated with the stimulation of these receptors [106,107,108]. In all cases, the observed activation of cholangiocyte proliferation is accompanied by an increase in cAMP levels [109]; inversely, the suppression of bile duct hyperplasia is associated with a decrease in cAMP levels [103].

A number of interleukins and growth factors are likely to participate in the regulation of cholangiocyte proliferation. In particular, IL1 and TNFa can stimulate the secretion of IL6 by cholangiocytes in a paracrine manner by stimulating p44/p42 MAPK activity (Figure 1) [110]. Other growth factors implicated as stimulators of cholangiocyte proliferation include EGF, FGF2, and TGFb2 [111].

### 3.2. Transdifferentiation Capacity of Cholangiocytes

Several successive stages are distinguished during the transdifferentiation of cholangiocytes into hepatocytes (Figure 2). Similar to hepatocyte transdifferentiation, there are cells that arise in the early stages of the process, as well as cells with an intermediate phenotype that appear in the later stages of transdifferentiation. The following marker proteins are present in all cells with an intermediate phenotype between cholangiocytes and hepatocytes: Alb, Hnf4a, and SOX9. Cells at different stages of transdifferentiation differ in their level of CK19 expression. As the hepatocyte phenotype develops, CK19 expression decreases and eventually disappears completely in mature hepatocytes [66,112].

In severe cases of chronic liver damage, the ductular reaction (DR) can be detected, morphologically presenting as an ectopic accumulation of cells that express cholangiocyte markers. The DR has long been considered as an example of the activation of liver progenitor cells residing in a dormant state within bile ducts [61,113,114]. The DR can be triggered by various manipulations involving hepatocytes. It has been shown that the knockout of the β-Catenin [115] or β1-integrin [116] gene in damaged livers leads to the proliferation of cholangiocytes, i.e., the DR, and their subsequent differentiation into hepatocytes.

Some authors distinguished several types of DR [114,117,118]. A type 1 DR (typical) is driven by the proliferation of cholangiocytes within the ducts of the portal triads. A type 2 DR (atypical) occurs through the differentiation of periportal hepatocytes into cholangiocytes. A type 3 DR arises from cells of the Hering canal [117,118]. The development of the DR is supported by the interaction of a complex network of cells—cholangiocytes, hepatocytes, neutrophils, tissue basophils, macrophages, lymphocytes, endothelial cells, and fibroblasts [61,114].

In humans, NCAM1 and TNFRSF12A serve as markers of the cholangiocytes involved in the DR [119]. TROP2 can also be used as a marker protein [120]. The forming clusters of cholangiocyte-like cells do not separate from the existing bile ducts and never lose their connection with them [121]. The involvement of hepatocytes in the DR cell progenitors is debatable. The contribution of such hepatocyte-derived cells probably depends on the nature of the injury. In cases where the biliary tree is primarily damaged, the transdifferentiation of hepatocytes is possible. In this case, SOX9+ hepatocytes are involved in the development of the DR [122]. However, cells formed by the division of existing cholangiocytes have a Spp1+ EpCAM+ phenotype and differ from Spp1+ EpCAM- cells originating from hepatocytes [63,121]. In chronic liver damage leading to the induction of DR, cholangiocytes that participate in DR formation differ in their capacity to proliferate. It has been demonstrated that cholangiocytes with a higher proliferative potential tend to be located in ducts of smaller diameter [121]. The cholangiocyte heterogeneity has also been noted under normal conditions [123].

Although the presence of liver progenitor cells in the biliary epithelium has been hardly considered until recently, another view of this problem has emerged. It is assumed that differentiated cholangiocytes are capable of a transition into so-called transient progenitor cells (CK19+HNF4α+), which are capable of differentiating into hepatocytes or returning to the cholangiocyte phenotype. This process is inhibited by NOTCH signaling and stimulated by the Wnt pathway (Figure 1) [112]. Recent data have emerged on the role of VEGFA signaling in cholangiocyte reprogramming [124]. Under conditions of toxic liver injury, the increased synthesis of VEGFA in the liver led to the appearance of a significant number of hepatocytes derived from cholangiocytes [124]. Previously, data were also obtained on the role of Hedgehog signaling in the development of the DR and the differentiation of liver cells with an intermediate phenotype between cholangiocytes and hepatocytes [125,126].

Other studies demonstrate that the ectopic SOX9+ clusters formed in DRs are virtually non-involved in the formation of new hepatocytes, but differentiate specifically into cholangiocytes. The number of hepatocytes is restored through their own proliferation [127]. The fate of SOX9+ hepatocytes derived from cholangiocytes may depend on metabolic nuclear receptors. PPARα activation promotes SOX9+ cell proliferation and differentiation into hepatocytes, while FXR activation has the opposite effect [128].

We have already mentioned the broader capabilities of liver epithelial cell transdifferentiation, particularly into pancreatic islet cells. In this regard, most studies have been focused on hepatocytes. However, a number of authors believe that cholangiocytes, expressing SOX9 in a similar manner to pancreatic duct cells, are more suitable candidates for this purpose. This approach was successfully confirmed by introducing the *Pdx1*, *Ngn3*, and *MafA* genes into a streptozotocin-induced diabetes animal model [129].

## 4. Conclusions

In summary, it can be concluded that hepatocytes and cholangiocytes have significant potential for proliferation and transdifferentiation during liver regeneration, which plays a key role in the repair process. Hepatocytes, as the main functional cells of the liver, show a high capacity to divide and transdifferentiate, activated by multiple signaling cascades. This capacity is crucial for maintaining homeostasis and normal functioning of the liver after resection. Cholangiocytes, in turn, also exhibit plasticity, which allows them to participate in the regeneration and restoration of bile ducts.

The examples discussed in this review highlight the importance of studying cellular plasticity in the context of liver regeneration. Understanding the mechanisms underlying the proliferation and transdifferentiation of hepatocytes and cholangiocytes may lead to the development of new treatments for liver diseases and improved outcomes after resection.

The pronounced plasticity of hepatocytes and cholangiocytes is of great clinical importance from the point of view of regenerative medicine. It is known that for many pathological conditions of the liver, the only treatment option is transplantation. An alternative to this procedure may be the transplantation of functionally active hepatocytes, which are also difficult to obtain from a pathologically altered liver. In this regard, a new source of hepatocytes may be cells obtained from cholangiocytes. This approach has already been tested in experimental studies [5,58].

Research in this area will provide insights into the molecular and cellular mechanisms that promote liver regeneration, which may ultimately contribute to the development of effective therapies for patients with liver failure and other liver diseases.

## Figures and Tables

**Figure 1 cells-14-01129-f001:**
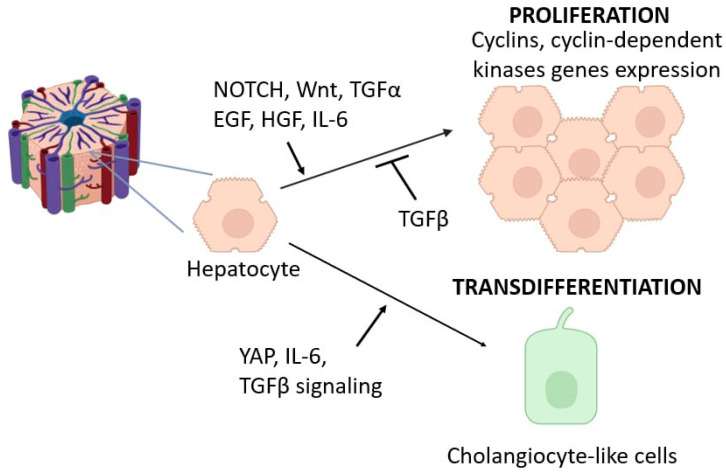
Manifestation of cellular plasticity in hepatocytes.

**Figure 2 cells-14-01129-f002:**
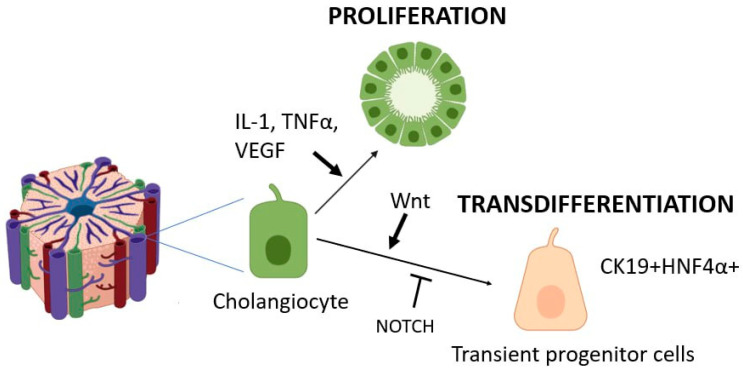
Manifestation of cellular plasticity in cholangiocytes.

## Data Availability

No new data were created or analyzed in this study. Data sharing is not applicable to this article.

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
