# Peer review of "Liver Regeneration as a Model for Studying Cellular Plasticity in Mammals: The Roles of Hepatocytes and Cholangiocytes"

_cells, 2025, doi:10.3390/cells14151129_

Round 1
Reviewer 1 Report
Comments and Suggestions for Authors
The authors provide a highly useful group of 110 reference covering all aspects of the processes described in the title of the article. This makes the article useful in a practical sense for future readers in this topic, since it is possible to find even opposing positions between authors of different articles in many of the articles referred to, as needed.
Two small syntactic issues need small change, as follows:
- The authors describe HGF as "found in large quantities in the connective tissue 103 matrix of the liver, especially..." HGF is found not in the connective tissue, but in the extracellular matrix between hepatocytes.
- In relation to TGFβ1, the authors state that it "stops the repair processes in the liver.." TGFβ1 is one of many factors that contribute to the termination of liver regeneration. A very important contribution of TGFβ1 is the formation of the new endothelial cells. Anoher important contributor to termination of liver regeneration is Integrin Linked Kinase.
Author Response
Reviewer #1:
The authors describe HGF as "found in large quantities in the connective tissue 103 matrix of the liver, especially..." HGF is found not in the connective tissue, but in the extracellular matrix between hepatocytes.
Thank you for your valuable comment. You are absolutely right. The text has been corrected.
In relation to TGFβ1, the authors state that it "stops the repair processes in the liver.." TGFβ1 is one of many factors that contribute to the termination of liver regeneration. A very important contribution of TGFβ1 is the formation of the new endothelial cells. Another important contributor to termination of liver regeneration is Integrin Linked Kinase.
Thank you for your valuable comment. Indeed, there are quite a few factors that stop liver regeneration. We wanted to emphasize the fact that TGFb has antiproliferative activity.
Reviewer 2 Report
Comments and Suggestions for Authors
Summary: This manuscript addresses a highly important and relevant topic: liver regeneration as a model for studying cellular plasticity, with a focus on hepatocytes and cholangiocytes. The subject matter holds significant interest for researchers in regenerative medicine and liver biology. While the topic itself is compelling, the current presentation of the review requires substantial revisions to meet the standards expected for publication.
Major Comments:
- Lack of Up-to-Date Content and Comprehensiveness: The review, despite its breadth, appears to be missing important recent advancements and key literature in the field of liver regeneration and cellular plasticity. For a comprehensive review, it is crucial that the authors incorporate the most current findings and ensure that the discussion reflects the cutting-edge understanding of the mechanisms involved. The current state suggests that the literature review is not sufficiently updated, which diminishes the "comprehensive" aspect.
- Introduction Length and Focus: As previously noted, the Introduction is excessively long and could be more concise. It should effectively set the stage for the review without delving into details that are better suited for subsequent, more in-depth sections. Condensing and streamlining this section would greatly improve the initial engagement for readers.
- Figure Quality and Informative Value: The figures included in the manuscript are currently of low quality. This applies to both their visual presentation (resolution, clarity of labels, aesthetic) and their ability to effectively convey information. Figures are critical for illustrating complex processes in a review article, and they need to be revised to be clear, high-resolution, informative, and professionally prepared. Authors should consider redrawing or significantly improving existing figures, and potentially adding new ones, to enhance the visual communication of their concepts.
Overall Writing Quality and Clarity: The manuscript's writing quality is consistently poor, which significantly hinders readability and the clear understanding of complex concepts. This goes beyond minor grammatical errors and includes issues with sentence structure, awkward phrasing, logical flow, and overall coherence. The language needs a thorough overhaul, ideally by a native English speaker with scientific expertise, to ensure clarity, precision, and professional presentation throughout the entire manuscript.
Author Response
Lack of Up-to-Date Content and Comprehensiveness: The review, despite its breadth, appears to be missing important recent advancements and key literature in the field of liver regeneration and cellular plasticity. For a comprehensive review, it is crucial that the authors incorporate the most current findings and ensure that the discussion reflects the cutting-edge understanding of the mechanisms involved. The current state suggests that the literature review is not sufficiently updated, which diminishes the "comprehensive" aspect.
Thank you for your valuable comment. You are absolutely right. The text has been corrected. Links to more up-to-date data have been added.
Introduction Length and Focus: As previously noted, the Introduction is excessively long and could be more concise. It should effectively set the stage for the review without delving into details that are better suited for subsequent, more in-depth sections. Condensing and streamlining this section would greatly improve the initial engagement for readers.
Thanks for the comments. Introduction shortened.
Figure Quality and Informative Value: The figures included in the manuscript are currently of low quality. This applies to both their visual presentation (resolution, clarity of labels, aesthetic) and their ability to effectively convey information. Figures are critical for illustrating complex processes in a review article, and they need to be revised to be clear, high-resolution, informative, and professionally prepared. Authors should consider redrawing or significantly improving existing figures, and potentially adding new ones, to enhance the visual communication of their concepts.
Thank you for your comments. The information provided in the review is now illustrated by two figures: separately for hepatocytes and cholangiocytes. We attach the figures separately to the submitted manuscript to maintain quality.
Overall Writing Quality and Clarity: The manuscript's writing quality is consistently poor, which significantly hinders readability and the clear understanding of complex concepts. This goes beyond minor grammatical errors and includes issues with sentence structure, awkward phrasing, logical flow, and overall coherence. The language needs a thorough overhaul, ideally by a native English speaker with scientific expertise, to ensure clarity, precision, and professional presentation throughout the entire manuscript.
Thank you for your comment. English has been corrected. Please find the attached certificate.

Round 2
Reviewer 2 Report
Comments and Suggestions for Authors The manuscript provides a clear, well-structured, and comprehensive review of current knowledge on liver regeneration, focusing particularly on the plasticity of hepatocytes and cholangiocytes. The topic is highly relevant, as understanding the cellular mechanisms of regeneration is key to developing novel therapeutic strategies for liver diseases. The authors successfully present the dual contribution of hepatocytes and cholangiocytes to liver regeneration in both homeostatic and injury contexts, and they offer a valuable synthesis of recent findings from experimental models and lineage tracing studies. The narrative is coherent, and the figures and references are appropriate. I recommend acceptance with minor revisions, as specified below.-
Title Formatting
Please revise the title to correct formatting and remove numbering artifacts (e.g., “2 CELLULAR PLASTICITY” and “3 HEPATOCYTES…”). A suggested title could be: Liver Regeneration as a Model for Studying Cellular Plasticity in Mammals: The Roles of Hepatocytes and Cholangiocytes. -
Clarify Terminology
Ensure that the terms “dedifferentiation,” “transdifferentiation,” and “bipotential progenitors” are clearly defined and consistently used throughout the manuscript to avoid confusion, particularly for readers less familiar with stem cell biology. -
Discussion Enhancement
The authors might consider briefly discussing how understanding hepatocyte/cholangiocyte plasticity can influence regenerative medicine or therapeutic approaches, particularly in chronic liver disease or transplantation contexts. -
Minor Language Polishing
A light proofreading is recommended to address occasional minor grammatical issues and to improve fluency and clarity, especially in longer paragraphs.
Author Response
Dear Reviewer, Thank you for your attention to our work and valuable advice. Below we have answered them step by step.
- Title Formatting
Please revise the title to correct formatting and remove numbering artifacts (e.g., “2
CELLULAR PLASTICITY” and “3 HEPATOCYTES…”). A suggested title could be: Liver Regeneration as a Model for Studying Cellular Plasticity in Mammals: The Rolesof Hepatocytes and Cholangiocytes.
Thanks for the suggestion, the title has been corrected.
- Clarify Terminology
Ensure that the terms “dedifferentiation,” “transdifferentiation,” and “bipotential progenitors” are clearly defined and consistently used throughout the manuscript to avoid confusion, particularly for readers less familiar with stem cell biology.
Thanks for your comment. The article has been updated accordingly.
- Discussion Enhancement
The authors might consider briefly discussing how understanding hepatocyte/cholangiocyte plasticity can influence regenerative medicine or therapeutic approaches, particularly in chronic liver disease or transplantation contexts.
Thanks for your comment. The article has been updated accordingly.
- Minor Language Polishing
A light proofreading is recommended to address occasional minor grammatical issues and to improve fluency and clarity, especially in longer paragraphs.
Thank you. Proofreading is done.
Sincerely, Dr. Andrey Elchaninov